# Distinct seasonal changes and precession forcing of surface and subsurface temperatures in the mid-latitudinal North Atlantic during the onset of the Late Pliocene

Xiaolei Pang[1,2], Antje H. L. Voelker[3,4], Sihua Lu[5], Xuan Ding[6]

[1]Institute of Ocean Research, Peking University, Beijing, 100871, China
[2]School of Earth and Spaces Sciences, Peking University, Beijing, 100871, China
[3]Instituto Português do Mar e da Atmosfera, Divisão de Geologia e Georecursos Marinhos, Av. Doutor Alfredo Magalhães Ramalho 6, 1495-165 Alges, Portugal
[4]Centro de Ciências do Mar do Algarve (CCMAR), Universidade do Algarve, Campus de Gambelas, Edf. 7, 8005-139 Faro, Portugal
[5]State Joint Key Laboratory of Environmental Simulation and Pollution Control, College of Environmental Sciences and Engineering, Peking University, Beijing, 100871, China
[6]School of Ocean Sciences, China University of Geosciences (Beijing), Beijing, 100083, China

*Correspondence to*: Xiaolei Pang (xiaolei.pang@pku.edu.cn)

**Abstract.** The Late Pliocene marks the intensification of Northern Hemisphere Glaciation, offering a unique opportunity to study climate evolution and ice-sheet related feedback mechanisms. In this study, we present high-resolution Mg/Ca-based sea surface (SST) and subsurface temperatures (SubT) derived from foraminiferal species *Globigerinoides ruber* and *Globorotalia hirsuta*, respectively, at the Integrated Ocean Drilling Project (IODP) Expedition 306 Site U1313 in the mid-latitudinal North Atlantic during the early Late Pliocene, 3.65 – 3.37 million years ago (Ma). We find distinct differences between our new *G. ruber* Mg/Ca-based SST record and previously published alkenone-based SST record from the same location. These discrepancies in both absolute values and variations highlight distinctly different seasonal influences on the proxies. The *G. ruber* Mg/Ca-based SST data, were primarily influenced by local summer insolation, showing a dominant precession cycle. Conversely, the variations in alkenone-based SST, dominated by the obliquity and lacking the precession cycle, are found to be more indicative of cold season changes, despite previous interpretations of these records as reflecting annual mean temperatures. A simultaneous decline in Mg/Ca-based SST and SubT records from 3.65 to 3.5 Ma suggests a diminished poleward oceanic heat transport, implying a weakening of the North Atlantic Current. A comparison with early Pleistocene *G. ruber* Mg/Ca-based SST records shows a shift in the dominant climatic cycle from precession to obliquity, alongside a marked increase in amplitude, indicating an enhanced influence of obliquity cycles correlated with the expansion of Northern Hemisphere ice sheets.

# 1 Introduction

The Late Pliocene, spanning from 3.6 to 2.58 million years ago (Ma), marks a pivotal transition in Earth's climate history. During this period, the climate shifted from a relatively stable and warm unipolar glaciated state to a cold and varied bipolar glaciated state associated with the development of the Northern Hemisphere Glaciation (NHG) (Lisiecki and Raymo, 2005; Westerhold et al., 2020). Marine sedimentary records of oxygen isotopes ($\delta^{18}O$) reveal that the onset of the NHG (oNHG) can be traced back to as early as ~ 3.6 Ma (Meyers and Hinnov, 2010; Mudelsee and Raymo, 2005). This is followed by an intensification of the NHG (iNHG) at ~2.7 Ma, characterized by a significant expansion of ice sheets, as evidenced by the abrupt increase of ice-rafted debris in marine sedimentary records (e.g., Bailey et al., 2013). From the iNHG onwards, the climate dynamics have been characterized by the glacial-interglacial fluctuations of the Northern Hemisphere ice sheets. The presence of ice sheets in the Northern Hemisphere has introduced powerful feedback mechanisms into the Earth's climate system (Meyers and Hinnov, 2010; Westerhold et al., 2020). The Late Pliocene thus provides an unique opportunity to study the characteristics of the climate systems with the involvement of ice-sheet related feedback mechanisms.

The mid- to high-latitude North Atlantic is the region most directly affected by the NHG. The North Atlantic Current (NAC), as the upper limb of the Atlantic Meridional Overturning Circulation (AMOC), currently serves as the primary source of oceanic heat for the high-latitude North Atlantic. It has been hypothesized that changes in the NAC's position or/and intensity may play a crucial role in both the oNHG and iNHG (e.g., Karas et al., 2020; Naafs et al., 2010). Sea surface temperature (SST) has been reconstructed across the northern North Atlantic to evaluate the influence from both the NAC and ice-sheets related feedbacks (Bolton et al., 2018; Karas et al., 2020; Lawrence et al., 2010; Naafs et al., 2020). Previously published Late Pliocene SST records from the mid- to high-latitude North Atlantic suggest warmer-than-present climate, with an overall cooling trend towards the Pleistocene (Naafs et al., 2020; Lawrence et al., 2010). On orbital timescale, SST changes are predominantly influenced by obliquity, with a notable absence of a significant precession cycle (McClymont et al., 2023).

There are a number of studies focusing on the Late Pliocene NAC changes (Karas et al., 2020; Naafs et al., 2010; Friedrich et al., 2013; De Schepper et al., 2009; Bolton et al., 2018; Hennissen et al., 2014). However, contrasting conclusions about NAC changes and its impacts are often drawn when analyzing SST records from various sites and even when examining SST records obtained from different paleotemperature proxies at the same site. For instance, the alkenone-based SST record from the Integrated Ocean Drilling Program (IODP) Site U1313 (41°00′N, 32°57′W) indicates a long-term decreasing trend from 2.9 to 2.5 Ma, with pronounced cooling during glacial periods. Combining alkenone-based productivity records, Naafs et al. (2010) proposed significant weakening of the NAC, especially during severe glacial times. They suggest that during these glacial periods, the NAC's flow direction likely shifted from its current northeastern trajectory to a more west-east direction. This would allow the subarctic front, the boundary between the subtropical (STG) and the subpolar gyre (SPG), to reach the vicinity of Site U1313. Yet, this conclusion is contested by subsequent research at the same Site U1313 (Bolton et al., 2018; Friedrich et al., 2013). Here, higher foraminifera *Globigerinoides ruber* white Mg/Ca-based SST indicate a generally warmer climate during the same time interval, with only modest cooling during glacial periods. This undermines the idea of a

substantially weakened NAC and the subarctic front being displaced into the mid-latitudes of Site U1313. Recently, based on a decreasing trend observed in *Globigerina bulloides* Mg/Ca-based SST at Site 610 (53°13′N, 18°53′W) between 3.65 to 3.5 Ma, Karas et al. (2020) proposed that a weakened NAC during this period preconditioned the climate for the oNHG. However, this conclusion is at odds with the warming trend recorded by alkenone-based SST records during the same period from nearby sites (Lawrence et al., 2009; Naafs et al., 2010).

The contrasting conclusions about the NAC changes based on the analysis of different SST records, might arise from the use of different proxies that could reflect different seasons. The alkenone-based SST, so far offering the most comprehensive and continuous records in the North Atlantic, are usually interpreted as representing annual mean temperature (Lawrence et al., 2010; Naafs et al., 2010, 2020). However, some studies from the mid-latitudinal North Atlantic suggest that alkenone-based SST are biased towards recording cold season conditions as a result of spring- or winter-time blooms of coccolithophorids (Bahr et al., 2023; Repschläger et al., 2023).

The foraminifera *G. ruber* white and *G. bulloides* are two commonly used species for Mg/Ca-based SST reconstruction in the North Atlantic. The *G. bulloides* Mg/Ca-based SST is usually viewed as reflecting an annual or spring signal (Hennissen et al., 2015; Karas et al., 2020). However, it might lean towards subsurface temperatures due to vertical migrations influenced by factors such as food supply and water column structure (McClymont et al., 2023). Recent research suggests that *G. bulloides* Mg/Ca may represent spring subsurface conditions (Repschläger et al., 2023). On the other hand, for the *G. ruber* white Mg/Ca-based SST, there is a general consensus in literature that it represents summer conditions in the mid-latitudinal North Atlantic (Bolton et al., 2018; Friedrich et al., 2013; Robinson et al., 2008).

Recent studies have underscored the importance of subsurface temperature (SubT) records, obtained from deep-dwelling foraminiferal species like *Globorotalia inflata* and *Globorotalia crassaformis*, in examining horizontal heat advection at depth (Bolton et al., 2018; Catunda et al., 2021; Reißig et al., 2019). In contrast to SST, which is highly responsive to seasonal forcing factors such as solar radiation and wind strength and direction, SubT exhibits minimal seasonal variability. This makes SubT a more reliable indicator for studying horizontal heat advection related to changes in ocean currents, free from seasonal interference. In the boundary regions of the North Atlantic STG, changes in SubT have been closely linked to the meridional movements of water mass of the STG (Bolton et al., 2018; Reißig et al., 2019). Given that the NAC transport occurs from the surface to depth (Daniault et al., 2016; Lozier, 2012), its variations should produce consistent effects in both SST and SubT. Therefore, the combined use of SubT and SST records could provide a more comprehensive identification of NAC-related variability, avoiding contrasting interpretation of NAC changes based on potentially seasonally biased SST records (e.g. Friedrich et al., 2013; Karas et al., 2020).

In this study, we provide high-resolution *G. ruber* white Mg/Ca-based SST and *Globorotalia hirstua* Mg/Ca-based SubT records for the oNHG period (3.65 – 3.37 Ma) at mid-latitudinal North Atlantic Site U1313 within the STG's northern region (sometimes also referred to as transitional waters; Fig. 1). By combining previously published alkenone-based SST data from the same location, we aim to gain a more detailed view of climate evolution associated with the NAC and the oNHG during the early Late Pliocene. We further compare *G. ruber* Mg/Ca-based SST and alkenone based SST from the early Late

Pliocene (3.65 – 3.37 Ma) with those from the Plio-Pleistocene transition (2.8 – 2.4 Ma) at Site U1313. This makes it possible to explore how the amplitude and dominant orbital cycle might have changed over time during the development of the NHG.

## 2. Material and methods

### 2.1 Sampling and Age model

IODP Site U1313 (Fig. 1), a re-drill of Deep Sea Drilling Program (DSDP) Site 607, was retrieved at a water depth of 3426 m from the base of the upper western flank of the Mid-Atlantic Ridge. Four holes were drilled at Site U1313 (Expedition 306 Scientists, 2006). The sediment samples analyzed in this study are from the primary shipboard splice of holes U1313B and U1313C, spanning the depth interval from 157.21 m to 172.14 m of the adjusted meter composite depth (amcd) scale (for details on the amcd see (Naafs et al., 2012); U1313B samples retain their original mcd depths). The 10 cc samples were taken over a 2 cm wide sediment interval at a spacing of 5 cm. In total, 323 samples were used for this study. Epibenthic foraminifera (*Cibicidoides wuellerstorfi*, *Cibicidoides mundulus*, and other *Cibicidoides spp.*) $\delta^{18}O$ were measured for those samples and integrated into the high-resolution benthic foraminiferal $\delta^{18}O$ record used to generate the age model of Naafs et al. (2020). That age model is based on tunning the U1313 $\delta^{18}O$ record to the global LR04 benthic $\delta^{18}O$ stack (Naafs et al., 2020). Accordingly, the time span corresponding to the investigated interval is estimated to be 3.65 to 3.37 Ma, with a temporal resolution of approximately 1 thousand years (kyr) for each sample.

### 2.2 Foraminiferal species and depth habitats

In order to establish temperature records from both surface and subsurface depth, this study utilized two species of planktonic foraminifera: *G. ruber* (white) and *G. hirsuta*. *G. ruber* white is a mixed layer species with a habitat depth ranging from 0 to 50 m (Anand et al., 2003), and it is widely used for reconstructing SST across the global ocean (e.g. Clark et al., 2024; Friedrich et al., 2013; Leduc et al., 2010; Lee et al., 2021; Medina-Elizalde and Lea, 2005; Pena et al., 2008). Notably, at Site U1313, previous studies have indicated that the *G. ruber* white Mg/Ca-based SST represent summer conditions (Bolton et al., 2018; Friedrich et al., 2013). On the other hand, *G. hirsuta* is a deep thermocline species with a habitat depth ranging from 0 to 800 m within the North Atlantic (Cléroux et al., 2013), but maximum abundances between 300 and 500 m in the vicinity of the Azores Islands (Rebotim et al., 2019). Its depth range, therefore, overlaps with *G. crassaformis*, which was used for reconstructing SubT at Site U1313 in the early Pleistocene (Catunda et al., 2021; Bolton et al., 2018). Because *G. crassaformis* was not being consistently observed within our study interval, persistently abundant *G. hirsuta* was chosen for reconstructing SubT instead.

### 2.3 Mg/Ca analysis and Paleotemperature reconstruction

For the trace element analyses, approximately 30 individual tests of *G. ruber* white and 20 of *G. hirsuta* were selected from the size fraction of 250 to 355 μm. The foraminiferal tests were gently crushed between two glass plates under a binocular microscope. The resulting crushed samples were subsequently transferred to 500 μl centrifuge tubes for further cleaning. The cleaning procedure adhered to the methods outlined in Barker et al. (2003) without using the reductive step. After cleaning, each sample was dissolved within a 0.075M $HNO_3$ solution followed by centrifugation to eliminate any residual particulate impurities. The resulting supernatant solution was transferred to a new 500 μl centrifuge tube. Measurements of Mg, Ca, Al,

and Mn concentrations were carried out using inductively coupled plasma mass spectrometry (ICP-MS, Bruker aurora M90) at Peking University. Al/Ca and Mn/Ca ratios were used for examining the cleaning efficiency. The Al values are usually lower than the detection limit of the ICP-MS. No anomalous Mg/Ca ratios were observed, nor was there any relationship between Mg/Ca and Mn/Ca ratios (Supplementary Fig. S1 and S2). Thus, no Mg/Ca data were rejected. The long-term reproducibility of Mg/Ca measurements, obtained by replicating analyses of a standard solution (Mg/Ca = 3.6 mmol/mol) along with samples over a four-month testing period, is 0.014 mmol/mol, corresponding to a relative standard deviation (RSD) of $\pm 0.4\%$ (1$\sigma$).

The residence time for Ca and Mg in the ocean are approximately 1 and 13 million years (Myr), respectively. Over the Cenozoic, the Mg/Ca ratio of seawater (Mg/Ca$_{sw}$) has exhibited notable secular variations (Zhou et al., 2021). Therefore, when using foraminiferal Mg/Ca ratios for reconstructing ocean temperatures for time periods exceeding 1 Myr, one needs to account for the influence of Mg/Ca$_{sw}$ variations (Evans and Müller, 2012). In order to address this concern, the measured Mg/Ca (Mg/Ca$_{meas}$) are corrected for the change in Mg/Ca$_{sw}$ (Zhou et al., 2021) by using the following relationship from Evans and Müller (2012): Mg/Ca$_{corr}$ = Mg/Ca$_{meas}$ $*\{(5.3/(Mg/Ca_{sw})\}^H$. The power coefficient H was set to 0.4 following the recommendation in the paper (Evans and Müller, 2012). It is worth noting that, considering that the duration of our study interval is less than 1 Myr, the applied correction does not induce alterations to trends or the relative temperature variations (Supplementary Fig. S3).

The corrected Mg/Ca ratios were converted to temperatures using the formula of Mg/Ca = 0.38±0.02 exp (0.09±0.003 *T) for *G. ruber* white (Anand et al., 2003) and the species-specific calibration Mg/Ca = 0.2±0.07 exp (0.18±0.057 T) for *G. hirsuta* (Cléroux et al., 2013). According to the chosen temperature calibration, correcting for the long-term evolution of Mg/Ca$_{sw}$ increased the *G. ruber*-based SST on average by ~0.8°C and the *G. hirsuta*-based SubT by ~0.3°C. The total uncertainties in reconstructed temperatures were estimated by propagating the uncertainties introduced by Mg/Ca measurements and temperature calibrations. The total uncertainties are on average about ±1 °C (1$\sigma$) for *G. ruber* Mg/Ca-based SST and ±4.7 °C (1$\sigma$) for *G. hirsuta* Mg/Ca-based SubT. The considerable uncertainty associated with *G. hirsuta* Mg/Ca-based SubT arises from the significant uncertainty in its calibration. This error is expected to be reduced with improvements in the calibration of *G. hirsuta*. For this study, the focus is more on the temporal variability and long-term trends of SubT itself, rather than comparing absolute temperature values with other temperature records. Therefore, for SubT records, the error in Mg/Ca measurements is more relevant in determining whether its own variability (such as orbital cycles and long-term trends) is significant. Considering only the measurement error, the corresponding *G. hirsuta* Mg/Ca-based SubT uncertainty is ±0.05 °C.

## 2.4 Spectral analysis

Spectral analysis was conducted by using the REDFIT module from the PAST4 software package (Hammer and Harper, 2001), the settings for the analysis are the same for all the records: Window = Welch, Oversampling = 3 and Segments = 3. For the purpose of filtering the data, the software package AnalySeries 2.0 was utilized (Paillard et al., 1996).

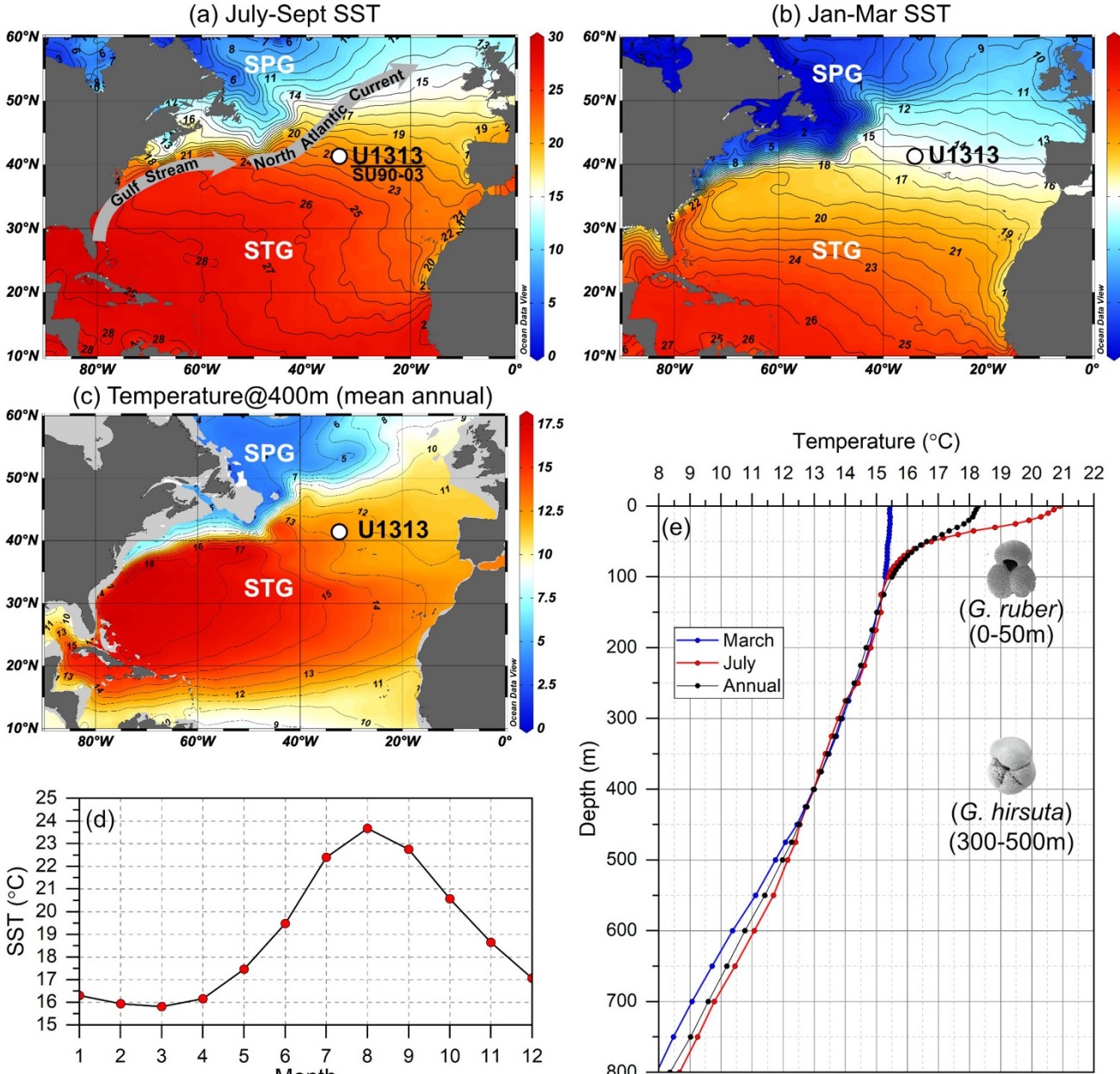

**Figure 1.** The location of Site U1313 and modern oceanographic settings. (a) Summer season (July to September) mean surface temperature. The main route of warm current Gulf Stream and its northern extension, the North Atlantic Current, are also presented. STG is subtropical gyre, SPG is subpolar gyre. Site SU90-03 (40.51ºN; 32.05ºW) which provides core top Mg/Ca ratios is presented by the same point of U1313 (41.00ºN; 32.96ºW) due to the proximity of locations. (b) Winter season (January to March) mean surface temperature and (c) mean annual water temperature at depth of 400 m. (d) Modern monthly mean surface temperature at site U1313, averaged from 2000 – 2015 derived from the Simple Ocean Data Assimilation (SODA) database (Carton and Giese, 2008). (e) Depth profile of temperature during different seasons

at site U1313, and the two studied planktonic foraminiferal species with their habitat depth indicated. Temperature data in (a), (b), (c) and (e) are sourced from the World Ocean Atlas 2013 (Locarnini et al., 2013).

## 3. Results

### 3.1 Mg/Ca and Temperature

The Mg/Ca ratios and resulting SST obtained for the *G. ruber* samples exhibit a range of 2.9 to 3.9 mmol/mol and 23.2 to 26.6 ℃, respectively (Fig. 2). The SST record exhibits small-scale oscillations over the course of the period. The highest value is observed at ~3.61 Ma, while the lowest value is approximately around 3.47 Ma. A distinct declining pattern is evident between 3.65 and 3.47 Ma.

As for the Mg/Ca ratios and the estimated SubT derived from the *G. hirsuta* samples, the values fluctuate within the range of 1.9 to 2.5 mmol/mol and 11.7 to 13.2 ℃, respectively. Notably, two distinct decreasing trends are observed in the SubT record. The first trend, similar to the SST record, spans from 3.65 to 3.5 Ma with a decrease of approximately 1℃, while the second trend commences from 3.45 Ma onwards, showing a decrease of about 0.5℃.

### 3.2 Spectral analysis results

Spectral analysis results for both *G. ruber* white Mg/Ca-based SST and *G. hirstua* Mg/Ca-based SubT records from this study (3.3 – 3.7 Ma) indicate prominent peaks at frequencies corresponding to the precession (19 – 23 kyr) and obliquity cycles (41kyr) (Fig. 3). The precession cycle is the predominant influence on the *G. ruber* white Mg/Ca-based SST, while the *G. hirsuta* Mg/Ca-based SubT is primarily governed by the obliquity cycle. Although the 100-kyr cycle appears in the spectral analysis results, it is not reliable as the short time span of the dataset (~ 300 kyr) does not allow for a statistically sound interpretation.

For further comparison, spectral analyses were also conducted on previously published alkenone-based SST records from the two intervals of 3.3 – 3.7 Ma (Fig. 3b) and 2.4 – 2.8 Ma (Fig. 3e) and on the *G. ruber* white Mg/Ca-based SST record from the younger period (2.4 – 2.8 Ma, Fig. 3d) at Site U1313. Together, these spectral analysis results highlight a significant observation: the alkenone-based SST and the Mg/Ca-based SST records exhibit distinct changes in the dominant periodic components during the NHG. Specifically, the alkenone-based SST record was consistently controlled by the obliquity throughout the transition from oNHG to iNHG, with a notable absence of a significant precession cycle. In contrast, the Mg/Ca-based SST record shows a shift in the dominant cycle from precession during the oNHG period (3.3 – 3.7 Ma) to obliquity during the iNHG period (2.4 – 2.8 Ma).

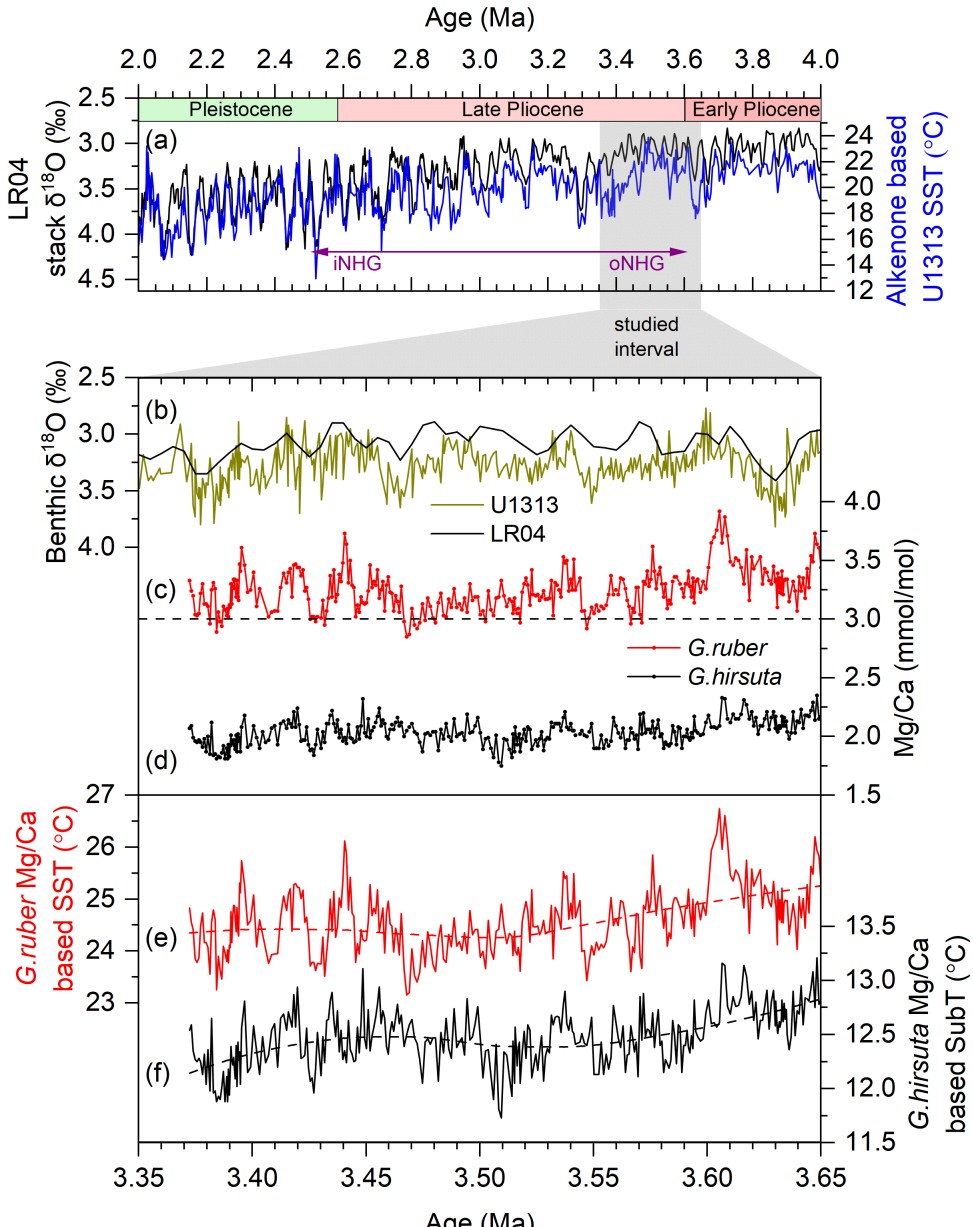

**Figure 2.** The studied interval (3.65 – 3.37 Ma) and paleoclimate data reconstructed from Site U1313. (a) The LR04 global benthic δ18O stack (Lisiecki and Raymo, 2005) and alkenone-based SST from Site U1313 (Naafs et al., 2020), the studied interval (3.65 – 3.35 Ma) is marked by the grey bar. (b) The LR04 benthic δ¹⁸O stack and benthic δ¹⁸O record from Site U1313, which has 0.64 per mil added to the original *Cibicidoides sp.* values (Naafs et al., 2020). (c) *G. ruber* Mg/Ca record, with the dashed line indicating the core top *G. ruber* white Mg/Ca value of 3 mmol/mol at site SU90-03. (d) *G. hirsuta* Mg/Ca record. (e) *G. ruber* white Mg/Ca-based SST record and (f) G. *hirsuta* Mg/Ca-based SubT record. Dashed line in (e) and (f) indicates smoothed curve by using Locally Estimated Scatterplot Smoothing (LOESS) method in R with a span of 0.75 and a polynomial degree of 2 to reveal underlying long-term trends.

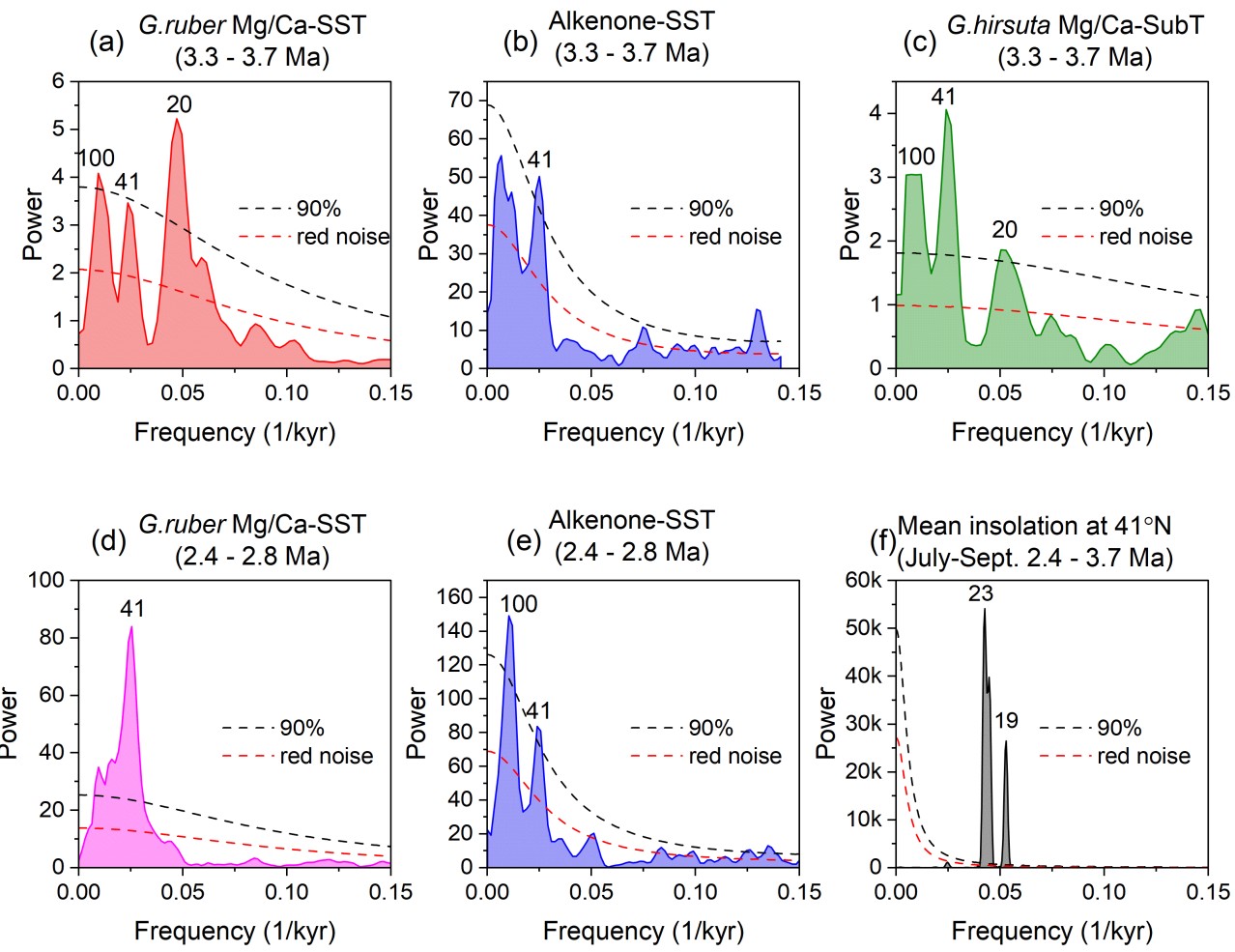

**Figure 3.** Spectral analysis results of paleoclimate records from Site U1313. *G. ruber* white Mg/Ca-based SST and *G. hirsuta* Mg/Ca-based SubT records in (a) and (c) are from this study. Alkenone-based SST records in (b) and (e) from Naafs et al. (2020). *G. ruber* white Mg/Ca-based SST records in (d) from Bolton et al. (2018) and Friedrich et al. (2013). (f) Mean insolation records are obtained from (Laskar et al., 2004). For details on the spectral analysis settings see subchapter 2.4.

## 4. Discussion

Our high-resolution Mg/Ca-based SST and SubT records provide new insights into climate dynamics of the mid-latitude North Atlantic during the early Late Pliocene, a period that marks the oNHG. The Mg/Ca-based SSTs exhibit distinct variations in both long-term trends and dominant orbital cycles compared to the previously published alkenone-based SST (Fig. 4), implying varying seasonal sensitivity of these proxies and thus seasonal SST changes in the mid-latitude North Atlantic in the early Late Pliocene. Our Mg/Ca-based SST and SubT records exhibit a simultaneous cooling trend from 3.65 to 3.5 Ma, contrasting with the warming trend observed in the alkenone-based SST records from the same period (Naafs et al., 2020), providing an alternate view on the meridional heat transport and associated NAC variations. On orbital timescales, we

observed a variety of dominant orbital cycles among Mg/Ca-based SST, SubT, and alkenone-based SST, indicating they are influenced by distinct mechanisms. Below, we focus on these critical features and discuss the driving forces behind them.

## 4.1 The warmth of mid-latitude North Atlantic in early Late Pliocene

The Late Pliocene is known for its global warmth, and is particularly pronounced in mid- to high-latitudes (Fedorov et al., 2013; McClymont et al., 2023). Previously reconstructed SST records spanning the study interval (3.65 – 3.35 Ma) in the mid- to high-latitude North Atlantic consistently indicate warmer-than-present temperatures (Karas et al., 2020; Lawrence et al., 2009; Naafs et al., 2010). Our new Mg/Ca-based SST confirms this Late Pliocene warmth.

The Mg/Ca-based SST consistently exceed the modern summer average of 23 °C (July-September, averaged from 2000-2015 AD) at Site U1313 (Fig. 4), suggesting a warmer-than-present mid-latitude North Atlantic in early Late Pliocene. To avoid the potential biases from the selection of Mg/Ca calibration and from the procedure of seawater Mg/Ca correction for this result, we further compared our *G. ruber* white Mg/Ca ratios with that of a core top sample from nearby core SU90-03 (40°30′N, 32°W; Fig. 1a, represented by same dot as U1313). The core top sample of SU90-03, estimated from the period of 0 – 4 ka, was non-reductively cleaned (the same method as used in this study) (Cléroux et al., 2008), thus no methodological bias occurs. Our *G. ruber* Mg/Ca ratios, whether or not adjusted for the secular variations of the seawater Mg/Ca, are generally above the value of 3 mmol/mol noted in the core top sample (Fig. 2c and Supplementary Fig. S3). Converting this core top *G. ruber* Mg/Ca ratio to temperature using the calibration selected in this study yields ~ 23 °C, which matches the modern summer temperature in this region. This supports the reliability of the selected calibration of Mg/Ca thermometry and reinforces the finding of the warmer-than-present summer SST at Site U1313 during the Late Pliocene.

What mechanism maintains the warmer SST at Site U1313 during the early Late Pliocene? The northward expansion of the STG's warm waters is likely the most important factor. The Pliocene is characterized by the substantial reduced meridional temperature gradient as clearly evidenced for the mid-Piacenzian Warm Period (Dowsett et al., 2012, 2016; Haywood et al., 2016). In the North Atlantic, the SST gradient from the equatorial to mid-latitude regions was up to 5°C smaller during the Pliocene compared to the Pleistocene (Fedorov et al., 2013; Dowsett et al., 2012), whereby the Pliocene SST increase is particularly pronounced in the mid- to high-latitudes (Fedorov et al., 2013; McClymont et al., 2023). The role of atmospheric $CO_2$ on the warmer climate remains uncertain due to the sparse paleo-$CO_2$ records available for the whole Pliocene and their significant uncertainties. Recently, the International Consortium of Cenozoic $CO_2$ Proxy Integration Project (CenCO2PIP) has vetted and synthesized the reliable paleo-$CO_2$ data available over the past 66 million years (CenCO2PIP Consortium et al., 2023). For our study interval (3.65 to 3.37 Ma), the reconstructed CenCO2PIP $CO_2$ concentration averaged around 300 ppm with a maximum value generally not exceeding 360 ppm (Fig. 4c). This is similar to the high-resolution reconstruction for 3.35-3.15 Ma, i.e. immediately following our study period (de la Vega et al., 2020). Notably, the $CO_2$ concentrations during the study interval in the Pliocene are overall lower than the modern value of 369 ppm in 2000 AD (Fig. 4b; Lan et al., 2024). In comparison, the reconstructed SST records in the study interval are warmer than the average temperature for the period 2000 - 2015 AD at Site U1313 (Fig. 4c). These observations indicate that $CO_2$ cannot solely explain the higher SST observed at Site U1313. This is in concert with the study by Fedorov et al. (2013), which showed that $CO_2$ is

not necessarily the primary driver for a warmer Pliocene climate. In summary, we conclude that the northward expansion of warm water from the STG is a primary process that maintained the warmer-than-present SST observed at Site U1313 during the early Late Pliocene.

**4.2 Distinct seasonal variations of SST during the early Late Pliocene: Comparing Mg/Ca- and alkenone-based SST**

Compared to the previously published alkenone-based SST from the same location, our new *G. ruber* white Mg/Ca-based SST record provides a different picture on the climate dynamics in the mid-latitudinal North Atlantic during the Late Pliocene. The *G. ruber* white Mg/Ca-based SST significantly differs from the alkenone-based SST in terms of both absolute values and changing patterns (Fig. 4c,d).

The Mg/Ca-based SST are overall higher than the alkenone-based SST, a pattern in line with previous observations at Site U1313 (Friedrich et al., 2013; Robinson et al., 2008). This difference can be attributed to the fact that *G. ruber* white Mg/Ca-based temperature reflects summer conditions (Repschläger et al., 2023; Robinson et al., 2008), whereas alkenone temperature represents annual mean or spring conditions (Naafs et al., 2020; Repschläger et al., 2023).

However, contrary to previous findings for the early Pleistocene that *G. ruber* white Mg/Ca and alkenone-based SST show generally synchronized variations (Bolton et al., 2018; Friedrich et al., 2013), our *G. ruber* white Mg/Ca-based SSTs display markedly different patterns in both the long-term trends and the dominant orbital cycles (Fig. 4c,d). The Mg/Ca-based SST shows a declining trend from 3.65 to 3.5 Ma, while the alkenone-based SST increases during the same period. Subsequent to 3.5 Ma, the alkenone-based SST notably decreases, while the Mg/Ca-based SST remains relatively stable.

A recent study from the vicinity of the Azores Islands to the south of Site U1313 suggests that the alkenone-based SST predominantly records spring temperatures (Repschläger et al., 2023). The alkenones are produced by coccolithophores that proliferate during spring blooms (March to May) in the mid-latitudinal North Atlantic (Cavaleiro et al., 2018; Lévy et al., 2005). As illustrated in Fig. 1d, March and April still fall within the coldest SST range at Site U1313. This evidence indicates that it is highly likely that the alkenone-based SST in our study area tends to reflect spring temperatures. However, it is important to note that we have no direct evidence to rule out the possibility that the alkenone-based SST represents the annual mean temperature, which is the calibration temperature used in the equation to convert the alkenone unsaturation index into SST (Müller et al., 1998; Naafs et al., 2010).

Compared to the uncertainty in interpreting absolute alkenone-based SST values for seasonal variations, we can draw more certain conclusions about the seasonal bias in its pattern of variability. Given our understanding that *G. ruber* Mg/Ca-SST reflects summer conditions, the distinctly different variability seen in the alkenone-based SST suggest that it reflects a season different from summer. Here, we refer to the alkenone-based SST as representing the cold season to contrast with the warm season indicated by *G. ruber* Mg/Ca SST, following Repschläger et al. (2023).

Therefore, the distinct different pattern of variability when comparing the Mg/Ca- and alkenone-based SST records at Site U1313 indicate that during the early Late Pliocene warm and cold season SST variations were influenced by different processes. The *G. ruber* white Mg/Ca-based SST is dominantly controlled by precession and shows a high correlation with local summer insolation (Fig. 5b). Presently located near the northern edge of the STG, the warmer-than-present *G. ruber*

white Mg/Ca-based SSTs during the early Late Pliocene imply that Site U1313 was consistently under the influence of the STG's warmer waters during the summer season. We propose that summer insolation forcing and associated STG variations are the primary drivers shaping the warm season *G. ruber* white Mg/Ca-based SST variations at Site U1313.

In contrast, the alkenone-based SSTs, dominated by obliquity with no sign of precession's influence (Fig. 3b), varied in phase with the benthic foraminiferal $\delta^{18}O$ record (Fig. 5c).   This correlation suggests that changes in alkenone-based SSTs are closely related to mechanisms and processes associated with ice dynamics. The ice-albedo effect is a crucial component of the climate system. For instance, in scenarios where the ice coverage expands, the surface albedo correspondingly increases. This results in more incoming solar radiation being reflected back into space, thereby cooling the surface temperatures. Such

a process could directly lower the SST at Site U1313 by the expansion of polar cold air masses towards lower latitudes. Moreover, cooling in the polar region would favor enhanced and southward-shifted westerlies (Bridges et al., 2023; Naafs et al., 2012). The strengthened westerly winds over Site U1313 would contribute to a decrease in SST by inducing excessive heat loss from the air-sea interface and by enhancing the mixing of upper waters (Fan et al., 2023). During the oNHG, despite the absence of large-scale ice sheets, the seasonal presence of thin (sea) ice in the cold season might have been significant (Clotten

et al., 2018; Knies et al., 2014). Changes in thin (sea) ice coverage would not have significantly impacted overall ice volume and thus would not have been reflected by the benthic $\delta^{18}O$ record, but their impact on the ice-albedo effect would have already been substantial. Furthermore, the reduction or disappearance of the thin (sea) ice during the warm season could explain why this high-latitude feedback did not dominate the Mg/Ca-based summer SST changes. Lawrence et al. (2009) proposed a similar mechanism to explain the unexpectedly high amplitude of alkenone-based SST observed in the high-latitude North Atlantic

during the early Pliocene warm period. We propose that ice-albedo effect and associated westerlies changes are the primary drives shaping the cold season alkenone-based SST variations at Site U1313.

          Inconsistencies between foraminiferal Mg/Ca- and alkenone-based SSTs are frequently observed and are mainly attributed to the different seasons and water depths they represent (Lawrence and Woodard, 2017; Leduc et al., 2010). Alkenone-based SST represents the upper 10 m of the water column (Herbert 2001), whereas foraminifera Mg/Ca-based SST

represents a greater depth range of the water column (Anand et al., 2003). For instance, the *G. ruber* white used in this study covers a range of 0-50 m (Anand et al., 2003). Moreover, the specific seasons and depths indicated by each proxy may vary across time and location (Leduc et al., 2010). In many cases from the Pleistocene, both proxy records exhibit similar variations on the glacial-interglacial timescale, and occasionally, they present similar values (Lawrence and Woodard, 2017; Lee et al., 2021). In such cases, these two proxies can be considered representative for comparable oceanographic conditions (Lawrence

and Woodard, 2017). For example, if the pattern of variability is comparable, one may expect that the signals reflect the same forcing responses, despite showing different absolute values (e.g. Lawrence and Woodard, 2017; Lee et al., 2021). However, in our case, the SST records provided by the two proxies are different in both variations and values, meaning that each proxy supplies information on different aspects of the climate system. We attribute these discrepancies in our study interval to the distinct seasonal variations at our site in the early Late Pliocene, a period when the Northern Hemisphere ice sheet related

feedback was not strong enough to synchronize changes in SSTs of both warm and cold seasons. This highlights the importance

of employing a multi-proxy approach to fully understand the characteristics of the climate system.

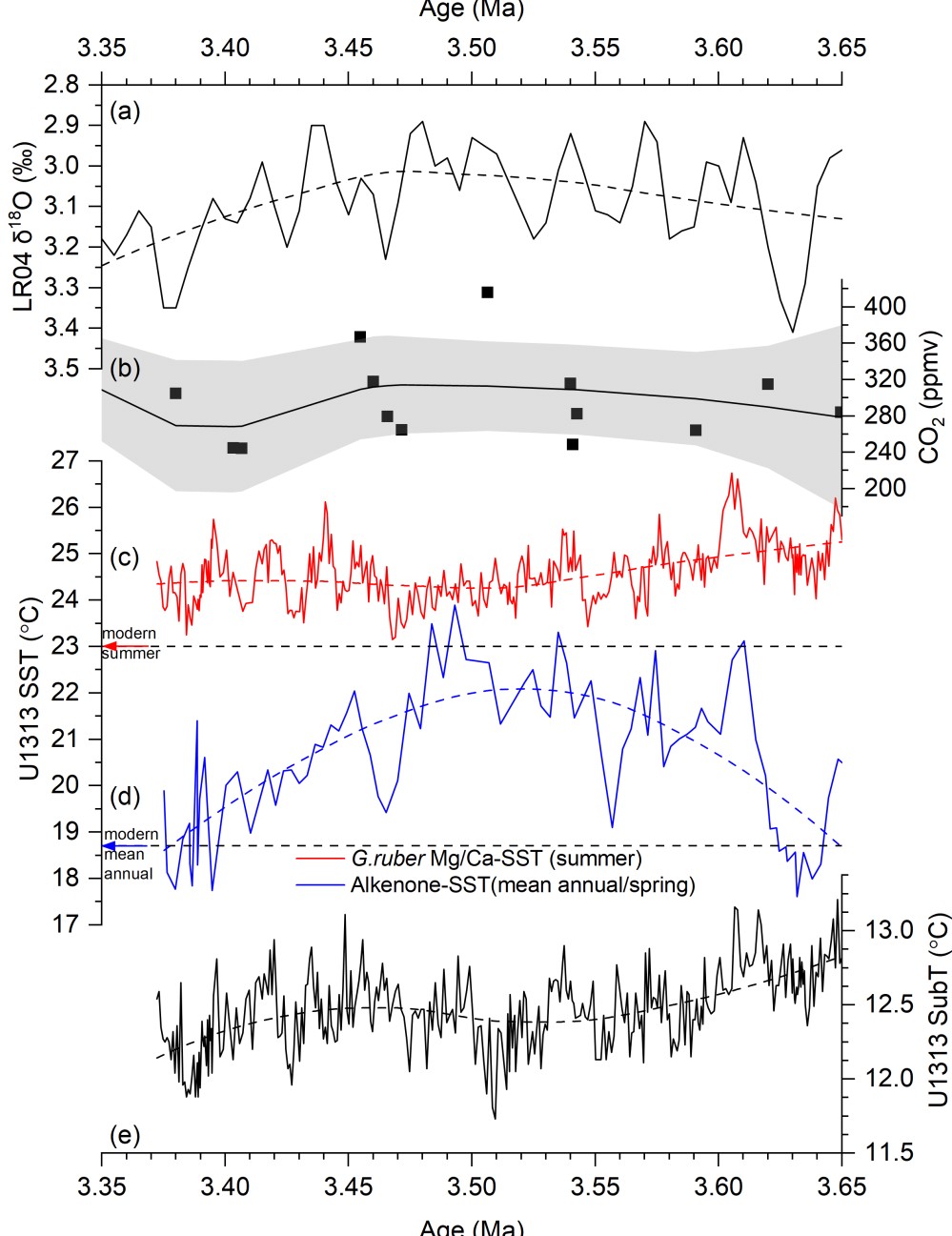

**Figure 4.** Distinct seasonal changes of SST records at Site U1313 during the early Late Pliocene. (a) The LR04 global benthic δ18O stack

(Lisiecki and Raymo, 2005). (b) Paleo-CO2 data available during the study interval (CenCO2PIP Consortium et al., 2023), with the black

line indicating the smoothed curve with 95% confidence interval . (c) *G. ruber* white Mg/Ca-based summer SST. (d) Alkenone-based SST

(Naafs et al., 2020). (e) *G. hirsuta* Mg/Ca-based SubT. Dashed lines in (a), (c), (d) and (e) indicate smoothed curve by using Locally Estimated Scatterplot Smoothing (LOESS) to reveal underlying long-term trends.

## 4.3 North Atlantic Current changes in the early Late Pliocene

The variations in poleward heat transport associated with the NAC has been considered important in influencing climate variations in the mid- to high-latitude North Atlantic, as well as in the evolution of the NHG (Karas et al., 2020; Naafs et al., 2010). Changes in SST records reconstructed from this region have commonly been linked to NAC variations (Bolton et al., 2018; Lawrence et al., 2009; Naafs et al., 2010). At our study location, Site U1313, situated near the modern northern boundary of the STG, the SST was considered particularly sensitive to northward heat transport and NAC variations (Bolton et al., 2018; Friedrich et al., 2013; Naafs et al., 2010).

The warmer-than-present SST observed at Site U1313 suggests a persistent northward extension of the STG during our study interval. This implies that the NAC consistently maintained a modern-like northeastern flow direction and high intensity during this period. Within this context, both Mg/Ca- and alkenone-based SST records show long-term trends, implying potential changes in the meridional heat transport. However, the contrasting trends observed between the Mg/Ca- and alkenone-based SST complicate a straightforward interpretation regarding NAC changes.

The northward water flow of the NAC occurs from the surface to depths exceeding 1000 meters (Daniault et al., 2016; Lozier, 2012). Temperature changes due to variations in lateral oceanic heat transport should manifest consistently at both surface and subsurface depths. The SST is influenced by a variety of factors, including significant seasonal fluctuations, which can obscure the signal of ocean current variations. In contrast, the subsurface waters inhabited by *G. hirsuta* (300 – 500m) display minimal seasonality (Fig. 1e), providing a clearer signal of long-term trends related to changes in lateral oceanic heat transport. Thus, by comparing SST and SubT records, we can better discern whether changes are due to variations in lateral oceanic heat transport. As shown in Fig. 4e, the Mg/Ca-based SubT exhibits a small but significant two-step cooling, each corresponding to the Mg/Ca- or alkenone-based SST records, reflecting two stages of possible NAC variation.

During the period between 3.65 and 3.5 Ma, the Mg/Ca-based SubT exhibited a cooling of about 1°C. Concurrently, the Mg/Ca-based summer SST showed a corresponding decrease of around 1.5°C (Fig. 4c and Fig. 5a). This consistent cooling trend in both surface and subsurface water depths provides strong evidence for the decrease in northward heat transport, implying the weakening of the NAC. This aligns with findings from a recent study which suggests a weakened NAC in response to a weaker AMOC strength during the same period (Karas et al., 2020). In contrast, the alkenone-based SST shows an approximate 4°C increase, implying that factors beyond the NAC lead to this warming. Notably, the alkenone-based SST appears to follow changes in benthic $\delta^{18}$O (Fig. 5c). From 3.65 to 3.5 Ma, the benthic $\delta^{18}$O data indicate a long-term decreasing trend in ice volume (Fig. 4a), implying a reduction in ice extent and the ice-albedo effect. We argue that the increasing trend in alkenone-based SST during this period, despite the weakening of the NAC, reflected a diminished influence of polar cold air masses and the westerlies in the cold season due to decreasing ice cover.

From 3.45 Ma onwards, the Mg/Ca-based SubT exhibits a second phase of cooling, but with a smaller temperature change (~ 0.5°C). The alkenone-based SST shows a clear decreasing trend (~ 4°C), but with an earlier onset at around 3.5 Ma

(Fig. 4d). This simultaneous cooling trend in both SubT and alkenone-based SST suggests a weakening of the NAC, which aligns with findings by Naafs et al. (2010). However, the Mg/Ca-based summer SST does not show a corresponding trend to the NAC change. Additionally, the decrease in SubT (~0.5°C) is smaller than the change observed in the 3.65-3.5 Ma period (~1°C). These results suggest that the NAC's influence during this stage was less significant than in the 3.65-3.5 Ma period. Therefore, the significant decrease in the alkenone-based cold season SST appears not to be solely explained by the weakening

NAC. The benthic $\delta^{18}O$ record indicates an increase in ice volume in the Northern Hemisphere during this period. We propose that, aside from the influence of a weakened NAC, the significant decrease in alkenone-based SST during this period was likely driven by mechanisms related to seasonal ice development, such as the southward expansion of polar cold air masses coupled with intensified westerly winds.

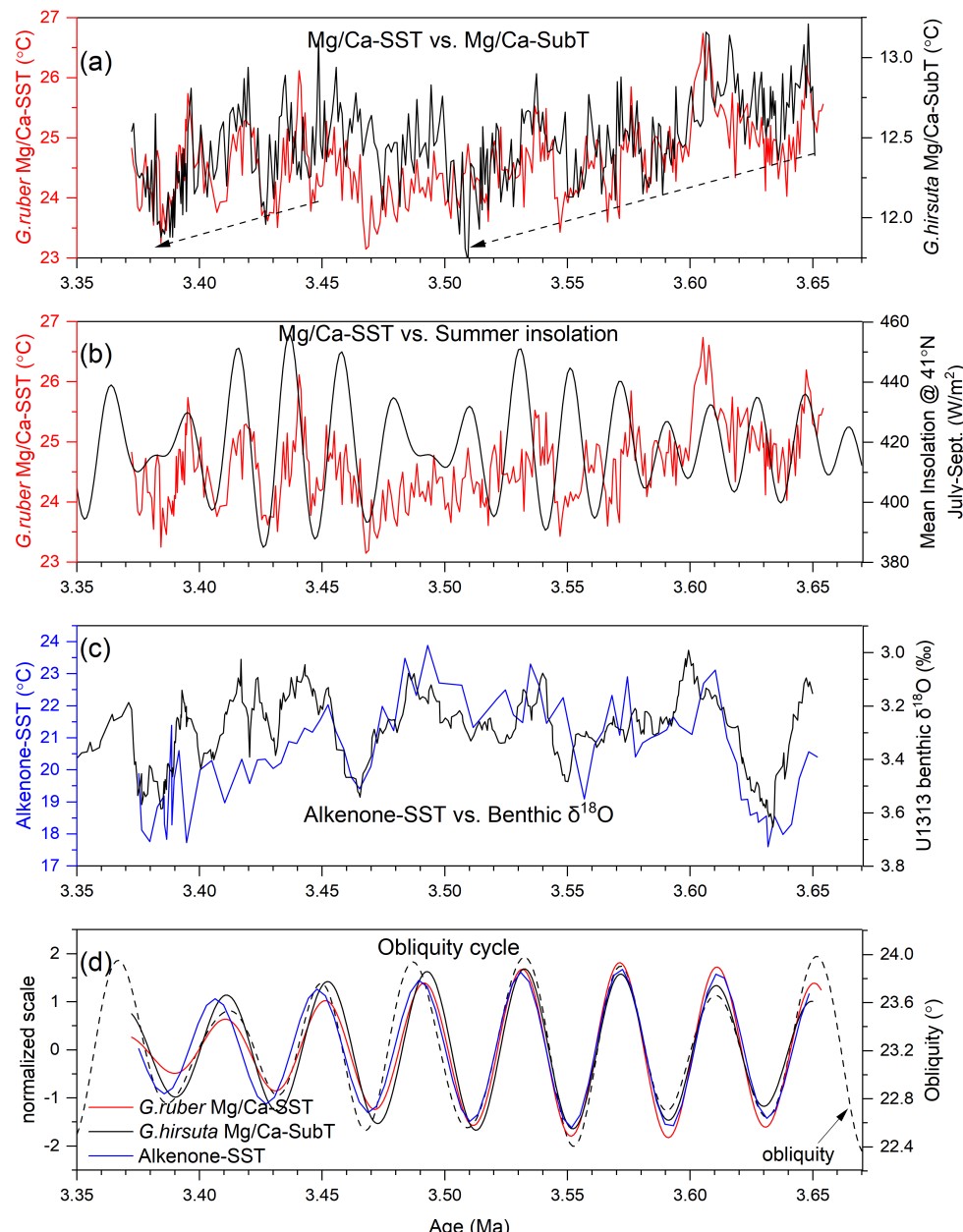

**Figure 5**. Comparison of *G. ruber* white Mg/Ca-based SST and *G. hirsuta* Mg/Ca-based SubT. (a) *G. ruber* white Mg/Ca-based SST (red) and *G. hirsuta* Mg/Ca-based SubT (black) records, with dashed arrows indicating long-term trends. (b) The comparison between *G. ruber* white Mg/Ca-based SST and mean insolation of July to September at 41ºN (Laskar et al., 2004). (c) Comparison between the LR04 benthic $\delta^{18}O$ stack (Lisiecki and Raymo, 2005) and the alkenone-based SST (Naafs et al., 2010). (d) Comparison between obliquity and filtered Mg/Ca-based SST, SubT and alkenone-based SST. Gaussian filtering, done with the software AnalySeries 2.0 (Paillard et al., 1996), centered

at the frequency of $0.025 \pm 0.003$, i.e., the obliquity band.

**4.4 From Precession to Obliquity: The Shifting Dominance in Late Pliocene – Early Pleistocene Sea Surface Temperature in the mid-latitudinal North Atlantic**

In this section, we explore the changes in the amplitude and pacing of climate in the mid-latitudinal North Atlantic through the NHG. Previous studies from the mid-to-high latitude North Atlantic were mostly based on alkenone-based SST records (e.g., Lawrence et al., 2009; McClymont et al., 2023; Naafs et al., 2010). The significant finding is that obliquity has consistently been the primary periodic component, with a lack of significant precession periods. Additionally, the amplitude did not show significant changes with the development of NHG. Combining the new Mg/Ca-based SST record from this study with those from the interval 2.8 to 2.4 Ma allows us to explore these changes from a summer perspective for the first time. The results suggest significant changes in both amplitude and dominant cycles through the NHG.

The *G. ruber* white Mg/Ca ratios from 2.8 – 2.4 Ma (Bolton et al., 2018; Friedrich et al., 2013) were corrected for seawater Mg/Ca changes and converted to temperature using the same calibration as in this study. The *G. ruber* white from that interval are of the *sensu stricto* morphotype (212-250 μm) with a calcification depth of 0-30 m, while this study includes both the *sensu stricto* and the *sensu lato* morphotypes (250 – 350 μm) with a calcification depth of 0 – 50 m (Anand et al., 2003). Friedrich et al. (2012) reported decrease in Mg/Ca with increasing test size for the species *G. ruber* white. Therefore, the use of the smaller sized *G. ruber* (s.s.) with narrower calcification depths is likely responsible for the higher (~ 1°C) interglacial SST values during the early Pleistocene compared to the warmer Late Pliocene (Fig. 6b). However, these differences are not expected to significantly affect the relative trends and periodicity in the resulting SST records.

When comparing the *G. ruber* white Mg/Ca-based SST records for our study interval and the Plio-/Pleistocene transition, two significant differences in terms of amplitude and periodicity emerge (Fig. 6). First, the amplitude of SST variations increases from approximately ~2 °C during the Late Pliocene to ~ 4 °C in the early Pleistocene. Second, there is a transition in the dominant cycle from precession to obliquity, with the precession cycle being absent between 2.8 – 2.4 Ma (Fig. 3d). Such a transition is not evident in the alkenone-based SST record (Fig. 3b,e).

During the early Late Pliocene (3.65 – 3.37 Ma), obliquity is the dominant cycle in both the alkenone-based SST and the *G. hirsuta* Mg/Ca-based SubT records, while being a secondary component in the *G. ruber* Mg/Ca-based SST records (Fig. 3). When filtering the three SST records for the obliquity band, synchronous variations are observed (Fig. 5d), suggesting a common process. As previously discussed in section 4.2, we suggest that the obliquity-induced thermal gradient and associated changes in the westerlies are the primary processes at play. During the oNHG, the seasonal ice coverage were significant to effect cold season SST, but it did not persist through the summer. This resulted in precession remaining the dominant cycle in the summer SST records.

During the 2.8 – 2.4 Ma interval, changes in both Mg/Ca- and alkenone-based SST records were aligned with the benthic $\delta^{18}$O record, indicating a dominant obliquity cycle and the absence of the precession cycle. Starting from the iNHG, ice sheets experienced significant growth during glacial periods compared to those of the early Late Pliocene, maintaining substantial volume even in summer. The expansion of the ice sheets enhanced the ice-related albedo effect, amplifying the meridional temperature gradients and winds. Studies indicate a significant glacial southward shift and intensification of

northern hemisphere westerlies beginning from 2.7 Ma (Abell et al., 2021; Naafs et al., 2012). From 2.6 Ma, an obvious southward reflection of the subarctic (subpolar) front occurred during glacial times (Bolton et al., 2018; Hennissen et al., 2014). The cold polar air and water masses, with intensified winds, periodically expanded towards the mid-latitudes, obliquity paced by the ice sheet dynamics (Bolton et al., 2018; Hennissen et al., 2014; Naafs et al., 2012). These processes contributed to severe cold temperatures at Site U1313 during glacial periods (Bolton et al., 2018; Naafs et al., 2010). As a consequence, the increased amplitude of temperature variations on the obliquity cycle overshadowed those of the precession cycle, causing the precession cycle to vanish from spectral analysis results. Therefore, obliquity transitioned from a secondary modulator of *G. ruber* white Mg/Ca-based SST during the early Late Pliocene to becoming the primary determinant in the early Pleistocene. These findings underscore ice volume fluctuations as the crucial factor driving the pace of orbital-scale climate variability in the mid-latitude North Atlantic.

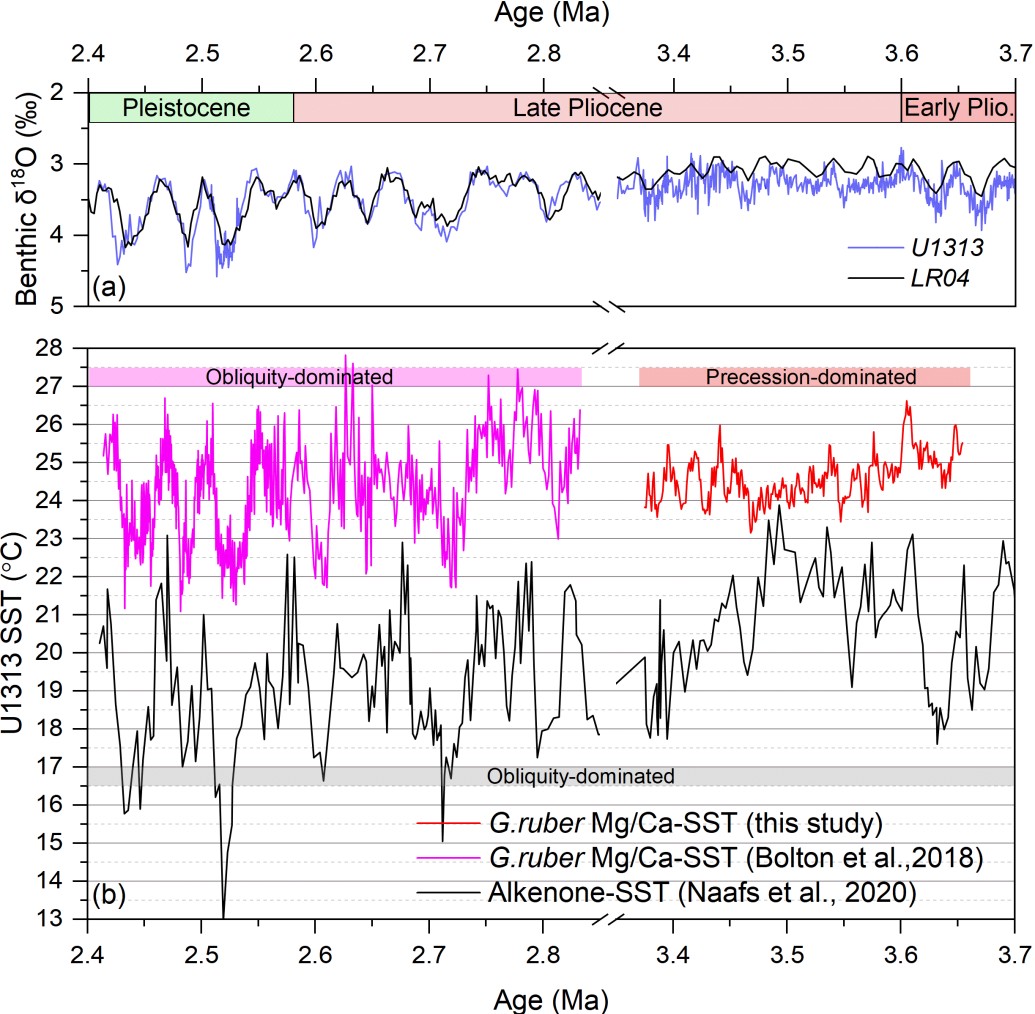

**Figure 6**. Comparison of the SST records between the Late Pliocene and Early Pleistocene. (a) Benthic $\delta^{18}$O of LR04 Stack (Lisiecki and Raymo, 2005) and Site U1313 (Naafs et al., 2020). (b) *G. ruber* white Mg/Ca-based SST (pink; Bolton et al., 2018; red, this study) and alkenone-based SST (black; Naafs et al., 2020)

## 5. Conclusions

In this study, we present high-resolution *G. ruber* white Mg/Ca-based SST and *G. hirsuta* Mg/Ca-based SubT records from the early Late Pliocene (3.65 – 3.37 Ma), coinciding with the oNHG, at IODP Site U1313 in the mid-latitude North Atlantic. The *G. ruber* white Mg/Ca-based SST, reflecting summer temperatures, were found to be higher by up to 3℃ compared to modern summer conditions at the same Site, indicating a consistent northward expansion of the warm STG waters during the early Late Pliocene.

When comparing our new *G. ruber* white Mg/Ca-based SST records with previously published alkenone-based SST records from the same Site, significant differences were noted in both the absolute values and the changing patterns. These discrepancies underscore the distinct seasonal sensitivities of the two proxies, revealing contrasting seasonal SST variations. Our findings suggest that the *G. ruber* white Mg/Ca-based SST are influenced by summer insolation as mirrored in the dominance of the precession cycle, whereas the obliquity dominated alkenone-based SSTs more likely reflect cold season

(spring) changes.

A simultaneous long-term decline in both Mg/Ca-based SST and SubT records from 3.65 to 3.5 Ma indicates a reduction in poleward heat transport, pointing to a weakening of the NAC. This observation lends support to the hypothesis that a reduced NAC may have played a role in initiating the Northern Hemisphere Glaciation (Karas et al., 2020).

Moreover, a comparison of the *G. ruber* Mg/Ca-based SST records from the Late Pliocene to the early Pleistocene

demonstrates significant shifts in the amplitude of temperature changes and in the dominant climatic cycles. The amplitude of SST variations increased from approximately 2°C in the Late Pliocene to about 4°C in the early Pleistocene, with a marked transition from the precession to the obliquity cycle. This change is attributed to ice-albedo feedback mechanisms and the progressive build-up of ice volume, which together significantly enhanced the influence of obliquity on climate dynamics.

**Data Availability.** The new data from this study are available at PANGAEA

(https://doi.pangaea.de/10.1594/PANGAEA.971263).

**Author contributions.** XP and AV initiated and designed the study. XP generated and analyzed Mg/Ca data with laboratory support from SL. XP prepared the manuscript with contributions from all co-authors.

**Competing interests.** The authors declare that they have no conflict of interest.

**Acknowledgements.** This research used samples provided by the Integrated Ocean Drilling Program (IODP). We greatly

thank the Bremen Core Repository personnel for collecting the samples (request 20250B of AV) and the student helpers/technicians in the Marine Geology department who washed the samples. We thank Liping Zhou for his insightful comments and facilitating laboratory support. AV received Portuguese national funds from FCT - Foundation for Science and Technology through projects UIDB/04326/2020 (DOI:10.54499/UIDB/04326/2020), UIDP/04326/2020 (DOI:10.54499/UIDP/04326/2020) and LA/P/0101/2020 (DOI:10.54499/LA/P/0101/2020).

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
