# Peer review of "Distinct seasonal changes and precession forcing of surface and subsurface temperatures in the mid-latitudinal North Atlantic during the onset of the Late Pliocene"

_EGUsphere, 2024_

## Author Comment (AC1)

**Response to Reviewer#1: Dr. Heather Ford**

In this manuscript, the authors use the Mg/Ca values of a surface dwelling and subsurface dwelling foraminifera to investigate climate in the North Atlantic during the early Late Pliocene. They find spectral and temperature differences from the existing alkenone-based record highlighting the seasonality in surface foraminifera and alkenones and the related climatological interpretations. I found the use of the subsurface dwelling records to reconstruct the North Atlantic current and poleward heat transporting compelling. I found the study well executed and the manuscript well-written. I have a few minor comments and suggestions to improve the clarify of the manuscript.

**We thank Dr. Heather Ford for her valuable comments and detailed suggestions, which have helped us improve the quality of the manuscript.**

**Below are our detailed point-by-point responses (in blue) to each comment (in black).**

**Please note that all line numbers mentioned in our responses correspond to the "No Markup" mode in the revised manuscript. Additionally, to prevent any errors or confusion with line numbers, for certain comments, we have directly copied and pasted the revised text into our responses.**

Line 122-123: write out abbreviations to full names.

To keep clear of this part, and in response to reviewer 2's comment regarding unnecessary content in this part, we have removed the beginning few sentences including the abbreviations.

Figure 1: Avoid rainbow color palette on figures. You can use BlueRed or Viridis in ODV

Followed the suggestion, the color scheme of Figure 1 has been updated to BlueRed in the revised manuscript

Figure 2: Update to ProbStack instead of LR04

As the age model for Site U1313 used in this manuscript was constructed by Naafs et al. (2020) by aligning U1313 benthic δ18O with LR04, it is more reasonable to plot U1313 benthic δ18O together with LR04 when needed, such as in Figure 2b and Figure 6a. If we replace LR04 with Prob-stack in Figure 2a, it will create an issue of having both LR04 and Prob-stack in this paper, which is unnecessary and might confuse the readers.

Despite some minor differences, LR04 and Prob-stack are highly consistent over the interval discussed in this paper (see the following figure). Considering that the benthic δ18O stack is mainly used as an indicator of polar ice volume changes in this paper, the choice between LR04 and the Prob-stack will not make significant difference. Additionally, the data comparisons in this paper are made within Site U1313 itself, thus independent of the age model.

Therefore, to avoid confusion and maintain consistency of the entire manuscript, we prefer to use only LR04 for all the figures.

[Figure]

Figure 3c: 100 peak blocked by 100 label – adjust position slightly

Although we also find it looks somewhat strange, this is the actual output of the spectral analysis and is not blocked.

Line 185: In the last sentence I would reiterative the precession cycle is absent from the 2.4 to 2.8 period, i.e. "The results indicate that the 2.4 to 2.8 Ma records are all dominated by obliquity, with a notable absence of a significant precession cycle after iNHG in comparison to the oNHG (3.3 to 3.7 Ma records)". The first time I read it and then looked at the figure I was confused so I would just be more specific.

We realize the lack of clarity in the last sentence. Following the suggestion, we have further revised it as follows (Line 188 to 193, revised manuscript):

"Together, these spectral analysis results highlight a significant observation: the alkenone-based SST and the Mg/Ca-based SST records exhibit distinct changes in the dominant periodic components during the NHG. Specifically, the alkenone-based SST record was consistently controlled by the obliquity throughout the transition from oNHG to iNHG, with a notable absence of a significant precession cycle. In contrast, the Mg/Ca-based SST record shows a shift in the dominant cycle from precession during the oNHG period (3.3 – 3.7 Ma) to obliquity during the iNHG period (2.4 – 2.8 Ma)."

Line 235-240: Do you mean 2000 AD?  I would re-write this as "For our study interval (3.65 to 3.37 Ma), the reconstructed CenCO2PIP CO2 concentration averaged around 300 ppm with a maximum value generally not exceeding 360 ppm (Fig. 4c).  This is similar to the high-resolution reconstruction for 3.35-3.15 Ma, i.e. immediately following our study period (de la Vega et al., 2020). For comparison, the modern CO2 value in 2000 was 360 ppm. Considering the reconstructed SSTs are warmer than the modern average temperature from 2000 to 2015 (Fig. 1d), the relatively higher CO2 could not be the primary cause of the warmer temperature during the late Pliocene."

We agree with the suggested rewrite, which improves the clarity of the text. We have accordingly made the revision in the revised manuscript (Line 242-247).

---

## Author Comment (AC2)

**Response to Reviewer#2**

Pang et al. present interesting new high-resolution foraminiferal Mg/Ca based SST and subsurface temperatures from the North Atlantic Site U1313 for the late Pliocene (3.65-3.4 Ma). The authors compare their data with previously published alkenone derived SST from the same Site and discover a different development. Especially, during 3.65-3.5 Ma, when alkenone derived SST showed a warming, their study suggests SST and subsurface cooling. They argue that this supports a previous study that suggested a weakening of the NAC during this time, which might have preconditioned the Northern Hemisphere Glaciation. The authors explain the differences between foraminiferal Mg/Ca based SST and alkenone derived SST due to representations of different seasons. They further argue that Mg/Ca derived SST experienced a marked change from precession dominated during the late Pliocene towards obliquity dominated during the Pleistocene. In contrast, alkenone SST were always driven by obliquity changes. I think the presented data is of interest in terms of a proxy comparison from an important core location when the Northern Hemisphere Glaciation begun. It is also a nice extension of previous published SST records in this region. The manuscript text needs some revisions at several locations when interpreting the results. Figures look good. In the following are my comments that should be addressed before publication:

**We thank the reviewer for their valuable comments and insights. Addressing these concerns has significantly improved the quality and clarity of the paper.**

**Below are our detailed point-by-point responses (in blue) to each comment (in black).**

**Please note that all line numbers mentioned in our responses correspond to the "No Markup" mode in the revised manuscript. Additionally, to prevent any errors or confusion with line numbers, for certain comments, we have directly copied and pasted the revised text into our responses.**

The authors explain the close similarity of alkenone SST from Site U1313 and global benthic $d^{18}O$ due to "ice-related albedo effects". However, the mechanisms are not sufficiently explained. Is a regional albedo effect meant? How does the albedo effect affect the temperature gradient at ~40ºN?

In the revised manuscript, we have added details in several places to improve the clarity of how changes in ice-albedo might influence SST variations at site U1313. The specific sections can be found in our responses to other similar comments below. Here, we present the first instance where we explain the related mechanism (In section 4.2, Line 288-294):

"The ice-albedo effect is a crucial component of the climate system. For instance, in scenarios where the ice coverage expands, the surface albedo correspondingly increases. This results in more incoming solar radiation being reflected back into space, thereby cooling the surface temperatures. Such a process could directly lower the SST at Site U1313 by the expansion of polar cold air masses towards lower latitudes. Moreover, cooling in the polar region would favor enhanced and southward-shifted westerlies (Bridges et al., 2023; Naafs et al., 2012). The strengthened westerly winds over Site U1313 would contribute to a decrease in SST by inducing excessive heat loss from the air-sea interface and by enhancing the mixing of upper waters (Fan et al., 2023)."

The authors argue that "summer" SST from *G. ruber* Mg/Ca in combination with subsurface temperature reconstructions are better suited to infer changes in NAC than "spring" SST from alkenones. However, I think the authors should also look at possible seasonal changes in NAC. Further, the connection between gyre circulation at 400 m water depth and NAC strength could be better explained.

The use of SubT aims to minimize the potential interference caused by seasonal variations in SST when discussing NAC changes. The seasonal variation's impact on SST may originate from changes in NAC or non-NAC factors, such as atmospheric circulation changes. Since NAC transport occurs from the surface to depths of more than 1000 meters, if the long-term changes in water temperature are due to NAC changes, we would expect to observe the same changing patterns in both SST and SubT. We did not presuppose that SubT and summer SST better reflect NAC changes; instead, we made judgments based on actual observations between SubT and SST, including both Mg/Ca-based and alkenone-based SST records.

The NAC originates from the STG. The record of changes in northward heat transportation at Site U1313 can be explained by both the meridional northern boundary changes and NAC variations. For example, increased northward heat transport indicates a poleward expansion of the northern boundary of the STG and a strengthened NAC. In this paper, we focus on discussing the NAC variations.

For better describing how combining use SST and SubT can better infer NAC changes we made several changes in the revised manuscript.

In the introduction section, between **lines 81-88**, we have revised the original content as follows: " Recent studies have underscored the importance of subsurface temperature (SubT) records, obtained from deep-dwelling foraminiferal species like *Globorotalia inflata* and *Globorotalia crassaformis*, in examining horizontal heat advection at depth (Bolton et al., 2018; Catunda et al., 2021; Reißig et al., 2019). Unlike SST, which is influenced by seasonal factors such as insolation and wind patterns, SubT exhibits minimal seasonal variability. This makes SubT a more reliable indicator for studying horizontal heat advection related to changes in ocean currents. In the boundary regions of the North Atlantic STG, changes in SubT have been closely linked to the meridional movements of water mass of the STG (Bolton et al., 2018; Reißig et al., 2019). Therefore, the combined use of SST and SubT can help distinguish the impacts of different climatic factors on their variations."

In the section 4.3, between **lines 339-346**, we have revised the orginal content as follows: "The northward water flow of the NAC occurs from the surface to depths exceeding 1000 meters (Daniault et al., 2016; Lozier, 2012). Temperature changes due to variations in lateral oceanic heat transport should manifest consistently at both surface and subsurface depths. The SST is influenced by a variety of factors, including significant seasonal fluctuations, which can obscure the signal of ocean current variations. In contrast, the subsurface waters inhabited by *G. hirsuta* (300 – 500m) display minimal seasonality (Fig. 1e), providing a clearer signal of long-term trends related to changes in lateral oceanic heat transport. Thus, by comparing SST and SubT records, we can better discern whether changes are due to variations in lateral oceanic heat transport. As shown in Fig. 4e, the Mg/Ca-based SubT exhibits a small but significant two-step cooling, each corresponding to the Mg/Ca- or alkenone-based SST records, reflecting two stages of possible NAC variation."

As mentioned above, subsurface Mg/Ca temperatures were used to support the surface temperatures, especially during 3.65-3.5 Ma. However, in the method section the authors show

an error in subsurface temperatures of about ±5°C, and further it is argued that this does not affect the results as the record is not used for comparison. I suggest revising this, better describing how subsurface temperatures can support SST.

We have revised the reason why the ±5°C does not affect the results in the method section (**Line 149-155**) as follows:

"The considerable uncertainty associated with *G. hirsuta* Mg/Ca-based SubT arises from the significant uncertainty in its calibration. This error is expected to be reduced with improvements in the calibration of *G. hirsuta*. For this study, the focus is more on the temporal variability and long-term trends of SubT itself, rather than comparing absolute temperature values with other temperature records. Therefore, for SubT records, the error in Mg/Ca measurements is more relevant in determining whether its own variability (such as orbital cycles and long-term trends) is significant. Considering only the measurement error, the corresponding *G. hirsuta* Mg/Ca-based SubT uncertainty is ±0.05 ℃."

More detailed comments in the manuscript:

Title: It might be better to focus on the differences between SST derived from foraminiferal Mg/Ca and alkenones. Why not mention the "onset of Northern Hemisphere Glaciation"?

Inconsistencies between Mg/Ca-SST and Alkenones-SST are commonly observed in previous studies, but the significant seasonal differences they reflect and the precession cycle observed in Mg/Ca-SST are the main new findings of this paper. Thus, we want to highlight these findings in the title. There is no consensus within the community regarding the timing of the onset of Northern Hemisphere glaciation. Therefore, such statements may imply different times to different readers. Thus, we prefer to provide clear time information for all readers by using 'the onset of the Late Pliocene' in the title. So, we prefer to keep the title as it is.

Line 24: Please include the influence of obliquity

Influence of obliquity on alkenone-SST has been added in the abstract (Line 23, revised manuscript)

Lines 31: Is "Cool-house climate" needed here? Maybe delete it as it may confuse the reader.

The "Cool-house climate" has been deleted, and this sentence has been further adjusted to (Line 32-33, revised manuscript):

"During this period, the climate shifted from a relatively stable and warm unipolar glaciated state to a cold and varied bipolar glaciated state associated with the development of the Northern Hemisphere Glaciation (NHG)"

Lines 85-87: This needs to be better described.

We have improved the whole paragraph for better express the important usage of SubT (Line 81-88, revised manuscript), We have previously addressed this issue in our response to above comment.

Line 112: Reference are needed for "is widely used".

References has been added (Line 111-112).

Lines 117-119: Please make two sentences out of this.

Have been changed (Line 117-119)

Lines 121-124: There is unnecessary information and repetition. Please shorten.

We have removed the unnecessary inforamtion in the begginnig of Section 2.3 (Line 121)

Line 136: Also provide absolute error.

The absolute error has been provided along with the Mg/Ca ratios of the standard resolution as follows(Line 130-133):

"The long-term reproducibility of Mg/Ca measurements, obtained by replicating analyses of a standard solution (Mg/Ca = 3.6 mmol/mol) along with samples over a four-month testing period, is 0.014 mmol/mol, corresponding to a relative standard deviation (RSD) of ±0.4% (1σ)."

Line 145: Please provide the reader with the difference between uncorrected and corrected Mg/Ca derived SST.

We have provided the difference between uncorrected and corrected Mg/Ca derived SST and SubT in the revised manuscript after introducing the calibration (Line 145-146):

"According to the chosen temperature calibration, correcting for the long-term evolution of $Mg/Ca_{sw}$ increased the *G. ruber*-based SST on average by ~0.8°C and the *G. hirsuta*-based SubT by ~0.3°C."

Line 160: Capital letter after a full stop.

We have corrected this in the revised manuscript.

Line 162: Provide latitudes and longitudes of the sites.

The latitudes and longitude of the sites have been provided in the caption of Figure 1.

Lines 174-176: Please indicate how much temperatures decrease

The corresponding temperature decrease has been added (Line 177-178):

"The first trend, similar to the SST record, spans from 3.65 to 3.5 Ma with a decrease of approximately 1°C, while the second trend commences from 3.45 Ma onwards, showing a decrease of about 0.5°C. "

Line 188: Indicate the time period.

Time period for the studied interval has been informed in the caption of Figure 2

Line 193: Give more information how the smooth was calculated

More inforamtion has been added about the smooth method in the caption of Figure 2

"Dashed line in (e) and (f) indicates smoothed curve by using Locally Estimated Scatterplot Smoothing (LOESS) method in R with a span of 0.75 and a polynomial degree of 2 to reveal underlying long-term trends."

Line 206: Please provide the reference for the alkenones

Correponding reference has been added (Line 214).

Line 235: Shorten the phrase. Is there something available with higher resolution?

The phrase has been shortened. In our knowledge, there is no higher resoltuion data available within our studied interval.

Line 251: The authors argue that alkenone derived SST indicate spring temperatures why mention here "mean or spring conditions"? This is confusing.

In our study, the difference between Mg/Ca- and alkenone-based SST is reflected in two aspects: absolute values and variation characteristics. Regarding absolute values, whether the alkenone-based SST represents spring temperatures or annual mean temperatures can be used to explain the differences with Mg/Ca-SST. In terms of variation characteristics, the alkenone-based SST must represent seasonal characteristics different from the summer characteristics of Mg/Ca-SST. In the initial draft, Line 251, we mentioned "mean or spring conditions" because, although we tend to believe that in our study interval the alkenone-based SST likely represents spring temperatures, we have no direct evidence to exclude the possibility that alkenone-based SST represents annual averages in terms of absolute values. In the revised manuscript, we have added few sentences to clarify our considerations regarding both absolute values and variation characteristics as follows (Line 264-278):

"A recent study from the vicinity of the Azores Islands to the south of Site U1313 suggests that the alkenone-based SST predominantly records spring temperatures (Repschläger et al., 2023). The alkenones are produced by coccolithophores that proliferate during spring blooms (March to May) in the mid-latitudinal North Atlantic (Cavaleiro et al., 2018; Lévy et al., 2005). As illustrated in Fig. 1d, March and April still fall within the coldest SST range at Site U1313. This evidence indicates that it is highly likely that the alkenone-based SST in our study area tends to reflect spring temperatures. However, it is important to note that we have no direct evidence to rule out the possibility that the alkenone-based SST represents the annual mean temperature, which is the calibration temperature used in the equation to convert the alkenone unsaturation index into SST (Müller et al., 1998; Naafs et al., 2010).

Compared to the uncertainty regarding its absolute values, we can draw more certain conclusions about the season represented by the variation characteristics of the alkenone-based SST. Given our understanding that *G. ruber* Mg/Ca-SST reflects summer conditions, the distinctly different changes in the alkenone-based SST must represent a season different from summer. In other words, whether the alkenone-SST in our study interval represents only spring or the annual mean in its values, its variation characteristics are certainly skewed towards a season opposite to the summer season indicated by the *G. ruber* Mg/Ca-based SST. Here, we refer to the alkenone-based SST as representing the cold season to contrast with the warm season indicated by *G. ruber* Mg/Ca SST."

Line 284: Define which depths are meant.

Depths information has been added in the revised manuscript as follows (Line 303-305):

"Alkenone-based SST represents the upper 10 m of the water column (Herbert 2001), whereas foraminifera Mg/Ca-based SST represents a greater depth range of the water column (Anand et al., 2003). For instance, the *G. ruber* white used in this study covers a range of 0-50 m (Anand et al., 2003)."

Lines 287-288: Unclear. Please better describe.

We have improved the describion in terms of the "interchangebly" in revised manuscript (Line 308-315) as follows:

"This means that these two proxies can be used interchangeably for estimating climate changes under certain conditions (Lawrence and Woodard, 2017). For example, in cases from the late Pleistocene period (e.g. Lawrence and Woodard, 2017; Lee et al., 2021), despite the absolute value differences between Mg/Ca- and alkenone-based SST, their variation characteristics often show roughly the same patterns on orbital to glacial-interglacial scales. In such situations, using either single proxy can yield the same conclusions about the SST variation characteristics and forcing mechanisms in the study area, allowing them to be used interchangeably. However, in our case, the SST records provided by the two proxies are different in both variations and values, meaning that using either single proxy would lead to different conclusions regarding temperature variation characteristics and controlling mechanisms, which means they cannot be used interchangeably."

Line 322: Replace "was" for "were".

The relevant sections have been revised, and this issue no longer exists.

Line 325: Better explain. See also main comment above.

We have provided a more detailed explanation of the mechanisms involved at the end of this paragraph (Line 354-356) as follows:

"We argue that the increasing trend in alkenone-based SST during this period, despite the weakening of the NAC, reflected a diminished influence of polar cold air masses and the westerlies in the cold season due to decreasing ice cover."

Line 331: Replace "decreasing" for "increasing"

The relevant sections have been revised, and this issue no longer exists.

Lines 331-333: Better explain. See also main comment above.

We have provided a more detailed explanation of the mechanisms involved at the end of this paragraph (Line 365-367) as follows:

"We propose that, aside from the influence of a weakened NAC, the significant decrease in alkenone-based SST during this period was likely driven by mechanisms related to seasonal sea ice development, such as the southward expansion of polar cold air masses coupled with intensified westerly winds."

Lines 343-346: Split sentence or shorten.

We have further polished this paragraph and the long sentence also be solved (First paragraph of Section 4.4, Line 377-383).

Lines 347-358: I suggest condensing this paragraph.

This paragraph has been condesed in the revised manuscript from 226 words down to 160 words (Line 384-391).

Line 355: Please indicate a value for "slightly higher"

We have added ~1°C for this difference in the revised manuscript (Line 389). We also deleted 'slightly,' as it is no longer appropriate in this context.

Lines 374-375. Please see main comments. The same mechanism is mentioned several times but not explained once in detail.

The mechanism has been better explained in detail in the revised manuscript; see the previous responses. For this specific section, we rearranged some sentences to make the explanation clearer (Line 407-416) as follows:
"The expansion of the ice sheets enhanced the ice-related albedo effect, amplifying the meridional temperature gradients and winds. Studies indicate a significant glacial southward shift and intensification of northern hemisphere westerlies beginning from 2.7 Ma (Abell et al., 2021; Naafs et al., 2012). From 2.6 Ma, an obvious southward reflection of the subarctic (subpolar) front occurred during glacial times (Bolton et al., 2018; Hennissen et al., 2014). The cold polar air and water masses, with intensified winds, periodically expanded towards the mid-latitudes, obliquity paced by the ice sheet dynamics (Bolton et al., 2018; Hennissen et al., 2014; Naafs et al., 2012). These processes contributed to severe cold temperatures at Site U1313 during glacial periods (Bolton et al., 2018; Naafs et al., 2010). As a consequence, the increased amplitude of temperature variations on the obliquity cycle overshadowed those of the precession cycle, causing the precession cycle to vanish from spectral analysis results. Therefore, obliquity transitioned from a secondary modulator of *G. ruber* white Mg/Ca-based SST during the early Late Pliocene to becoming the primary determinant in the early Pleistocene."

---

## Author Response (AR2)

We appreciate the constructive comments and detailed suggestions, which have helped us improve the quality of the manuscript.

Below are our detailed point-by-point responses (in blue) to each comment (in black).

Please note that all line numbers mentioned in our responses correspond to the "Track change verison" in the revised manuscript. Additionally, to prevent any errors or confusion with line numbers, for certain comments, we have directly copied and pasted the revised text into our responses.

**Page 1**
Line 21: distinctly different seasonal influences on the proxies.
We have updated this sentence following the suggestion (line 21).

Line 22:
We agree with the suggestion and have deleted the section (line 22).

**Page 3**
Line 85: Is the effects of winds solely restricted to the ocean surface? Consider to delete "and wind patterns"

We aim to express that on a seasonal scale, winds are a major factor leading to seasonal temperature differences. In comparison, subsurface waters are less responsive to these seasonal changes, especially in open ocean areas rather than upwelling regions. Because wind patterns are considered an important factor in generating significant seasonal differences in this study, we have chosen to retain this term. We have rephrased the sentence to improve the clarity as follows (Line 83-84):

"In contrast to SST, which is highly responsive to seasonal forcing factors such as solar radiation and wind patterns, SubT exhibits minimal seasonal variability."

Line 89: ...different climatic factors on their variations. - unclear, phrasing. Please rephrase to clarity and be more specific

The corresponding sentence has been rephrased for improving the clarity and be more specific as follows (Line 88-91):

"Given that the NAC transport occurs from the surface to depth (Daniault et al., 2016; Lozier, 2012), its variations should produce consistent effects in both SST and SubT. Therefore, the combined use of SubT and SST records could provide a more comprehensive identification of NAC-related changing signals, avoiding potential contrasting interpretation of NAC changes based solely on the seasonally biased SST records (e.g. Friedrich et al., 2013; Karas et al., 2020)."

**Page 4**
Line 102: re-drill
We have made the suggested change.

Line 103: Ridge. Four holes…

We have made the suggested change.

Line 107: which species?
The specific species now be listed (Line 110).

Line 108: measured
We have made the suggested change.

**Page 5**
Line 161: If the error of the calibration is so large and you don't really relate to the absolute temperatures, would it be better to only show the raw Mg/Ca data rather than the temperatures?

Although we don't relate to the absolute temperatures, the SubT's relatively change, which is independent to the error of the calibration, is an important information when discussing the NAC changes in Section 4.3.

**Page 7**
Line 186:
We have deleted it.

**Page 10-11**
Line 255: I would keep the full name, but maybe the abbreviation is not needed since its given in the reference? I know the reviewer asked for a shortening here, but abbreviations should be defined.

We now include both the full name and the abbreviation when it is first introduced (Line 247).

Line 261: year 2000 AD
We have made the suggested change.

Line 262: year 2015 to 2000 AD
We have made the suggested change.

Line 262: may not be?
Due to the different time scales considered when comparing the overall warmer long term Pliocene responses and the modern, can you be certain that CO2 have no effect, or could it be that the SSTs are still adjusting to the forcing?

We realized that the previous expression about the relationship between CO2 and SST at Site U1313 is unclear and may misleading the effect of CO2 forcing. We have make corresponding changes as follow (Line 251-261).

"Notably, the $CO_2$ concentrations during the study interval in Pliocene are overall lower than the modern value of 369 ppm in 2000 AD (Fig. 4b; Lan et al., 2024). In comparison, the reconstructed SST records at Site U1313 are warmer than the average temperature for the period 2000 - 2015 AD (Fig. 4c). These observations indicate that $CO_2$ cannot solely explain the higher SST observed at Site U1313. This is in concert with the study by Fedorov et al. (2013), which showed that $CO_2$ is not necessarily the primary driver for a warmer Pliocene climate. In summary, we conclude that the northward expansion of warm water from STG is a

primary process that maintained the warmer-than-present SST observed at Site U1313 during the early Late Pliocene."

**Page 11-13**
Line 289: ... by comparing the pattern of variability between the alkenone and Mg/Ca based SST.

We realized the related sentence was unclear. While we did not use the exact wording suggested, we have rephrased the sentence to improve clarity. It now reads as follows (Line 284- 286):

"Compared to the uncertainty in interpreting absolute alkenone-based SST values for seasonal variations, we can draw more certain conclusions about the seasonal bias in its pattern of variability."

Line 290: variability seen
We have made the suggested change.

Line 290: suggests that the alkenone data represents a season different from summer.
We have made the suggested change.

Line 291-293:
We have made the suggested change.

Line 294: , following Repenschläger et al. (2023).
We have made the suggested change.

Line 297: distinct different pattern of variability when comparing the Mg/Ca- and alkenone-based SST records...
We have made the suggested change.

Line 307: ice dynamics as in changes in ice volume/extent? or related to an assumption that when precession were lower you have larger ice sheets and more sea ice due to colder conditions?

Ice dynamics here refer to changes in ice volume and extent. We think the extent is more important than the volume when discussing the ice-albedo effect. For example, thin ice extent on the sea and land may not be large enough to significantly change the volume (i.e. the benthic d18O ), but it is already sufficient to impact the ice-albedo effect.

Line 311: Even though there are traces of Pliocene sea ice, would it be enough to shift the westerlies far enough south to significantly impact U1313? With a glacial type Nordic Seas sea ice extent the southwesterlies are shifted southwards and in a more westerly direction, but how much of an effect can be expected in a Pliocene climate with limited sea ice and smaller ice sheets (and from the PlioMIP simulations a northward shift of the westerlies are suggested). Consider to acknowledge the uncertainties involved.

Considering the modern center of the westerlies is currently between approximately 30 and 60 ºN, it is unlikely that the location of U1313 (41ºN) was outside the range of the westerlies during our study interval, despite potential northward shifts of the westerlies during the warm Pliocene. Thus, any changes in the westerlies are able to impact the Site U1313.

The effect of ice albedo should be limited in summer, but its winter time variations can still have a significant impact on the SST of the mid- to high-latitude North Atlantic. An evidence supporting this come from Site 982 (58 ºN), where alkenone-based SST variability during the warm Pliocene is comparable to the glacial-interglacial variability of the Pleistocene. Lawrence et al. (2009) proposed same explanation related to the ice feedback mechanisms.

Further supporting evidence comes from the long-term evolution of SST and δ18O relationships, especially when we view it from the young to old age. As illustrated in Figure 6, during the early Pleistocene, both SST proxies exhibit synchronous changes with global ice volume (δ18O), displaying a clear obliquity cycle. This synchronicity suggests that ice-related feedback mechanisms, such as the southward shift of colder air and water masses during glacial times, impact both the summer and winter SST at Site U1313. Returning to our study interval in the warmer Pliocene, alkenone-based SST still maintains a synchronous relationship with δ18O, displaying an obliquity-dominated cycle, suggesting that the same ice-related feedback mechanisms are still effective during the cold season.

Line 332: Suggested rephrasing: In such cases, these two proxies can be considered representative for comparable oceanographic conditions. (ref).
We have made the suggested change.

Line 334-335: Unclear sentence. Suggested rephrasing:
For Example, if the pattern of variability is comparable one may expect that the signals reflect the same forcing responses, despite showing different absolute values.
We have made the suggested change.

Line 338: suggested rephrasing: ... meaning that each proxy provide information on different aspects of the climate system.
We have made the suggested change.

**Page 15**
Line 384: add reference to this statement.
I do not see how changes in sea ice should impact the benthic d18O record? Please clarify the statement.

We have rephrased the related sentence. There is no longer a need for a specific reference. We now use "ice extent" instead of "sea ice," and the implication that sea ice should impact the benthic δ18O record has been removed. It now reads as follows:

"From 3.65 to 3.5 Ma, the benthic δ18O indicate a long-term decreasing trend in ice volume (Fig. 4a), implying a reduction in ice extent and the ice-albedo effect."

**Page 16**
Line 393: The meaning of the last part of this sentence is not clear. Line 394: why/based on what? Line 395: role for what

We have rephrased the related sentences to solve the above several concerns. They now reads as follows (Line 376-380):

"However, the Mg/Ca-based summer SST does not show a corresponding trend to the NAC change. Additionally, the decrease in SubT (~0.5°C) is smaller than the change observed in the 3.65-3.5 Ma period (~1°C). These results suggest that the NAC's influence during this stage was less significant than in the 3.65-3.5 Ma period. Therefore, the significant decrease in the alkenone-based cold season SST appears not to be solely explained by the weakening NAC."

Line 399: volume
We have made the suggested change.

Line 404: could U1313 rather be impacted by the interplay between the STG and SPG? likes to the comment on how large (or small) the seasonal sea ice cover is likely to have been at this time.

Unlikely. The interplay between the STG and SPG would not have directly influenced the temperature variations at U1313 during our study interval. At present, U1313 is located near the northern boundary of the STG. In the studied interval, both alkenone and Mg/Ca-SST show warmer-than-present values, implying a northward extension of the STG. As a result, U1313 was closer to, and probably within, the STG. The NAC likely maintained a northeastward flow similar to the present, isolating the STG from the SPG. It is suggested that the first significant glacial incursion of subarctic front surface water above Site U1313 did not occur until ~2.6 Ma (Bolton et al., 2018).

However, this does not negate the significant influence of the ice-albedo effect on U1313 by altering atmospheric circulation, particularly during the cold season. For a detailed explanation, please refer to our previous response.

---

## Author Response (AR3)

Dear Dr. Risebrobakken,

Thank you very much for accepting our manuscript for publication.

In response to your comment regarding lines 307-310, we have rephrased the text to avoid the implication that the ice sheet is a seasonal feature. We have also clarified our discussion on ice dynamics and the ice-albedo effect to align with the points raised in our response letter. Below is the revised version (lines 298-303):

"During the oNHG, despite the absence of large-scale ice sheets, the seasonal presence of thin (sea) ice in the cold season might have been significant (Clotten et al., 2018; Knies et al., 2014). Changes in thin (sea) ice coverage would not have significantly impacted overall ice volume and thus would not have been reflected by the benthic $\delta^{18}O$ record, but their impact on the ice-albedo effect would have already been substantial. Furthermore, the reduction or disappearance of the thin (sea) ice during the warm season could explain why this high-latitude feedback did not dominate the Mg/Ca-based summer SST changes."

Regarding the data availability, the data submitted to PANGAEA has been approved and published. As this process included editorial review, we have retained the PANGAEA link and removed the Zenodo link from the manuscript.

Thank you for your guidance and for the opportunity to publish in Climate of the Past.

Best regards,
Xiaolei Pang and co-authors